# Carbon Nanotube-Mediated Delivery of PTEN Variants: In Vitro Antitumor Activity in Breast Cancer Cells

**DOI:** 10.3390/molecules29122785

**Published:** 2024-06-11

**Authors:** Rigini M. Papi, Konstantinos S. Tasioulis, Petros V. Kechagioglou, Maria A. Papaioannou, Eleftherios G. Andriotis, Dimitrios A. Kyriakidis

**Affiliations:** 1Laboratory of Biochemistry, Department of Chemistry, Aristotle University of Thessaloniki, 541 24 Thessaloniki, Greece; kstasioul@chem.auth.gr (K.S.T.); petran_k@hotmail.gr (P.V.K.); kyr@chem.auth.gr (D.A.K.); 2Laboratory of Biological Chemistry, School of Medicine, Aristotle University of Thessaloniki, 541 24 Thessaloniki, Greece; mpapaioannou@auth.gr; 3Laboratory of Organic Chemical Technology, Department of Chemistry, Aristotle University of Thessaloniki, 541 24 Thessaloniki, Greece; e.andriotis@yahoo.gr

**Keywords:** PTEN, multiwalled carbon nanotubes, protein delivery, breast cancer cells, T-47D, MCF-7, ZR-75-1, cytotoxicity, proliferation, apoptosis

## Abstract

Phosphatase and tensin homologue deleted on chromosome 10 (PTEN) is a crucial tumor suppressor protein with frequent mutations and alterations. Although protein therapeutics are already integral to numerous medical fields, their potential remains nascent. This study aimed to investigate the impact of stable, unphosphorylated recombinant human full-length PTEN and its truncated variants, regarding their tumor suppression activity with multiwalled-carbon nanotubes (MW-CNTs) as vehicles for their delivery in breast cancer cells (T-47D, ZR-75-1, and MCF-7). The cloning, overexpression, and purification of PTEN variants were achieved from *E. coli*, followed by successful binding to CNTs. Cell incubation with protein-functionalized CNTs revealed that the full-length PTEN-CNTs significantly inhibited cancer cell growth and stimulated apoptosis in ZR-75-1 and MCF-7 cells, while truncated PTEN fragments on CNTs had a lesser effect. The N-terminal fragment, despite possessing the active site, did not have the same effect as the full length PTEN, emphasizing the necessity of interaction with the C2 domain in the C-terminal tail. Our findings highlight the efficacy of full-length PTEN in inhibiting cancer growth and inducing apoptosis through the alteration of the expression levels of key apoptotic markers. In addition, the utilization of carbon nanotubes as a potent PTEN protein delivery system provides valuable insights for future applications in in vivo models and clinical studies.

## 1. Introduction

PTEN (Phosphatase and tensin homolog deleted on chromosome 10) is a ubiquitously, well-characterized dual specificity lipid and protein phosphatase. PTEN mutations and the loss of its activity are frequent in solid and hematological malignancies, rendering the *PTEN* gene among the most altered human tumor suppressor genes [1]. The absence or alteration of PTEN activity has been related to many primary and metastatic malignancies, and to a therapeutic interest in the era of personalized medicine [2,3,4]. The tumor-suppressing function of PTEN relies on its ability to decrease the augmented concentrations of phosphatidylinositol-3,4,5-triphosphate (PIP3), which are produced by the phosphoinositide-3 lipid kinases (PI3’Ks) [5,6,7,8]. PIP3 serves as a critical lipid secondary messenger in the process of tumorigenesis, activating signaling molecules involved in various cellular processes, including survival, proliferation, cell motility, and invasion [8,9,10]. Mutations in the *PTEN* gene or its promoter are widespread, at a rate close to p53. The inherited mutations in PTEN are linked to certain cancer predisposition syndromes, such as Cowden syndrome (CS) with a prevalence of approximately 80%, and Bannayan–Riley–Ruvalcaba syndrome (BRRS) with an estimated frequency of 60% [11,12,13,14]. Somatic mutations in only one allele and a loss of heterozygosity are more often observed in glioblastoma and endometrial carcinoma, while mutations in both alleles occur during cancer progression in the brain, prostate, breast, endometrial cancer, and melanoma, with approximately 50% incidence [15].

The structure of PTEN is characterized by five distinct functional domains: the phosphatidylinositol-4,5-bisphosphate binding domain (PDB), the protein tyrosine phosphatase domain (PTP), which harbors the active site, the type II calcium-independent (C2) domain, the carboxyl-terminal tail, and the PDZ domain. The PDZ domain is crucial for mediating protein–protein interactions and plays a pivotal role in tethering receptor proteins in the membrane to cytoskeletal components [8]. The N-terminal domain possesses the active site capable of interacting with phosphoinositide substrates, where most mutations are localized. The association of PTEN with the cytoplasmic membrane is one of the most difficult events in PTEN-associated cancers [5,8]. It is known that the stable phosphorylated form of PTEN is present at higher levels in healthy, non-cancerous cells and must be converted to the unphosphorylated active form to release the active site, yet this form is prone to degradation [16,17]. The C-terminal domain of PTEN plays a critical role in maintaining and regulating PTEN stability [18,19]. Additionally, the PDZ domains of PTEN have been reported to exert a significant impact on its capacity to suppress the PI3K/AKT pathway [20].

Several efforts have been conducted to reactivate or restore PTEN function in cell lines or animal models. An interesting approach came from Ai and his colleagues [21], where after the downregulation identification of PTEN and its downstream target CDKN1B, they established a recombinant adeno-associated virus (rAAV) expressing PTEN and CDKN1B, and delivered it to transgenic adenocarcinoma mouse prostate (TRAMP) mice. Their results indicated that PTEN overexpression is a crucial event for cell migration and cell-cycle progression inhibition [21]. In parallel, apoptosis was stimulated, the lifespan of TRAMP mice was extended, and the growth rate of tumor xenografts was aborted [21]. The coexpression of TP53, BIM and PTEN remodeled and altered the cancer signaling network in vitro and improved the therapeutic effect in non-small-cell lung cancer (NSCLC) in vivo [22]. Their results were quite promising, since the apoptotic cell percentage reached 94.9%, while in the mouse models, except for the strong inhibitory effect, side effects were not detected [22]. Members of the same team have previously reported similar effects with only the PTEN gene, indicating that the restoration expression of PTEN and the simultaneous inactivation of the PI3K-AKT-mTOR axis successfully repressed cell proliferation and induced apoptosis [23]. In the same field of prostate cancer, a codelivery approach of AKT3 siRNA and PTEN plasmid by antioxidant cerium oxide nanoparticles (CeNPs) could penetrate prostate cancer cells, restore PTEN expression and knockdown AKT3 expression [24]. The same advantageous nanosystem conferred DNA damage and apoptosis in prostate cancer cells and provided fundamental insights into its effectiveness against prostate cancer [24]. A smart and solid methodology was introduced by Kim and colleagues [25], where they designed lipid nanoparticles with enhanced selectivity against PD-L1 overexpressing cells both in vitro and in vivo. The effective mRNA delivery of PTEN with the above-mentioned nanoparticles promoted autophagy-mediated immunogenic cell death in 4T1 tumors, leading to potent anticancer immune response [25]. In the field of glioblastoma, where the BBB penetration represents one of its challenges [26], Liu and colleagues managed to develop a nanoparticle platform for PTEN mRNA delivery that significantly improved tumor inhibition [27]. Lin and colleagues studied PTEN restoration in *PTEN*-mutated and *PTEN*-null cell lines through PTEN mRNA delivery in nanoparticles [28]. The same research team concluded that the combinatorial effort of PTEN mRNA delivery with an anti–programmed death–1 antibody yields a highly potent antitumor effect in both a subcutaneous *PTEN*-mutated melanoma model and an orthotopic *PTEN*-null prostate cancer model. Additionally, the combined treatment induces immunological memory in the *PTEN*-null prostate cancer model [28].

Numerous studies have indicated that a wide range of materials, including liposomes, polymers, and carbon nanotubes (CNTs), can be utilized to stabilize and deliver proteins for potential biomedical applications [29]. Moreover, several therapeutic methods have emerged in the past few years to address cancer cells that exhibit reduced PTEN activity. These studies employed various delivery systems for either the *PTEN* gene or its encoded protein, such as virus-based systems [30,31], gelatin hydrogel microspheres [32], exosome-based systems [33], magnetic iron oxide nanoparticles [34], aerosol-based systems [35] as well as PTEN transduction to various human cancer cells [36,37,38].

CNTs have been widely utilized for their theranostic potential [39] owing to their unique needle-like shape, which allows them to conjugate or adsorb onto diverse types of biomolecules [40]. Multiwalled CNTs consist of multiple layers of graphene sheets over one layer, with a diameter ranging from 2 to 50 nm [41]. Among their advantages, CNTs exert a prominent status due to their large surface, small size, invasiveness of lipid membranes, exceptionally high drug loading capacity and potential to conjugate various compounds with medicinal and diagnostic value [42,43,44,45]. It has been described that the CNTs’ internalization mechanism into cells is mainly endocytosis [46,47]. However, the selective suppression of endocytosis pathways revealed that many mechanisms are involved in the internalization of CNTs, including micropinocytosis, caveolae-mediated endocytosis, and clathrin-dependent endocytosis [48]. The passive diffusion of CNTs through the lipid bilayer can also be achieved regardless of temperature or endocytosis [49].

In addition, the efficacy of CNTs in breast cancer therapy has been demonstrated over time by numerous studies of the in vitro and in vivo delivery of genes, shRNA, proteins, and several drugs [50,51,52]. In a related study, Lay and coworkers developed single-walled PEGylated CNTs loaded with paclitaxel (PTX), which effectively killed MCF-7 breast cancer cells with IC_50_ lower than free PTX, indicating a potential therapeutic advantage and a promising approach [53].

In our previous study, we examined heterozygosity at the 10q23 locus and identified mutations in exons 1, 5, 7, and 9 in patients diagnosed with breast carcinoma or precursor lesions [4]. Furthermore, we assessed PTEN protein expression in both systemic circulation and corresponding tissue specimens from these patients. Our findings indicated that breast cancer development is often correlated with a high incidence of PTEN mutations, leading to the truncation of the protein and the subsequent loss of its activity, rather than a loss of expression [4]. Notably, this study demonstrated that peripheral blood analysis can provide an effective method to identify *PTEN* mutations and estimate PTEN protein expression in human cancer [4]. Various studies have focused on different methods to restore PTEN function in PTEN-deficient cancer cell lines. Protein therapy [13,14,54], gene therapy [11,12,55], and the use of PTEN-Long protein as an exogenous agent [56] have been shown to trigger cell cycle arrest, apoptosis, and tumor cell death with respective challenges for each approach.

The objective of this study was to investigate the impact of recombinant human full-length and truncated variants of PTEN regarding its tumor suppression activity. Another goal of this study is to develop a prospective delivery system for the functional, stable and unphosphorylated PTEN protein into breast cancer cells. The cloning, overexpression, and isolation of the full-length recombinant human PTEN (hPTEN1) and two truncated fragments containing the N-terminal (hPTEN2) and C-terminal (hPTEN3) domains, respectively, were accomplished in the *Escherichia coli*. To evaluate the biological effect of the conjugated CNTs with hPTEN1 and its two truncated fragments, we determined their function on the viability of T-47D breast cancer cells and apoptosis induction in the PTEN-deficient ZR-75-1 cells, as well as in the non-PTEN deficient breast cancer cell line, MCF-7. The conjugation of PTEN onto PEGylated multiwalled CNTs appears to significantly elevate the expression levels of crucial pro-apoptotic proteins, contributing to the promotion of programmed cell death. Simultaneously, there is a noticeable reduction in the levels of key anti-apoptotic proteins, further emphasizing the potential therapeutic efficacy of this immobilization technique. The inherent tumor-suppressive properties of PTEN in conjunction with the unique features of CNTs are a promising tool, and further exploration may yield valuable insights into developing targeted and effective therapies for combating cancer.

## 2. Results

### 2.1. Cloning, Expression, and Isolation of Recombinant hPTEN1 and Its Two Truncated Fragments

Three recombinant proteins were overexpressed and purified as C-terminal hexa-histidine fusions, the full-length of human PTEN (hPTEN1), and two truncated fragments of the protein. The first truncated version contained the N-terminal region, consisting of the active site and the TI loop (hPTEN2), while the second one contained the C-terminal region, consisting of the CBR3 loop, the PEST sequences, and the PDZ region (hPTEN3), with affinity to the membrane lipids.

The open reading frame of human PTEN was amplified using the recombinant vector pORF9-*hPTEN* v21 (InvivoGen, San Diego, CA, USA) as a template, to express hPTEN1 and its two truncated fragments, as described in Materials and Methods, Section 4.3. The sequences of the inserts from the selected colonies were confirmed by sequence analysis (VBC Biotech Service Gmbh, Wien, Austria). The pET-*hpten1*, pET-*hpten2* and pET-*hpten3* constructed vectors were then transformed into *E. coli* BL21 (DE3) competent cells. Induction with IPTG was performed as described in Materials and Methods. The total cells from 1 mL of bacterial cell culture, before and after the induction, were collected and electrophoresed on a 10% (*w*/*v*) SDS polyacrylamide gel. The desired fusion proteins were detected and immunostained with rabbit polyclonal anti-His-Tag antibody (Appendix A).

Furthermore, the purification of the overexpressed proteins was performed. Intense bands of hPTEN1, hPTEN2 and hPTEN3 were displayed in the expected range of MW at the fraction of inclusion bodies, suggesting that the overexpression of all three proteins resulted in the formation of inclusion bodies. Therefore, further purification of all proteins with affinity chromatography and Ni-TED columns was performed in the presence of 8 M urea. The eluted proteins were refolded with gradual removal of urea by dialysis, as described in Section 4.4. The purity of all eluted proteins was estimated by SDS-PAGE and silver staining of the gels (Appendix A), thus proving that the purification processes were successful. The isolation of the recombinant proteins, after denaturation and protein refolding, was performed at the percentage of 79% for hPTEN1, 93% for hPTEN2 and 80% for hPTEN3. This analysis was carried out by using ImageJ 1.53t software (https://imagej.net/software/imagej/, accessed on 13 November 2023).

### 2.2. Immobilization of Recombinant hPTEN1 and Its Two Truncated Fragments onto Carbon Nanotubes

The immobilization of the highly purified, recombinant proteins hPTEN1, hPTEN2 and hPTEN3 onto CNTs aims to facilitate their import into cancer cells in a stable form, while retaining their activities. Owing to its maintained capacity and diffusion within cells, as well as their ease of cell membrane penetration, CNTs find a variety of applications in biological systems. For this reason, oxidized CNTs (9.5 nm diameter × 1.5 nm length) were employed in this study (Sigma-Aldrich Chemie GmbH, Steinheim, Germany). An external alteration of CNTs with bis(3-aminopropyl)polyethylene glycol (PEG) has the potential to convert them into more hydrophilic, non-cytotoxic, and deterrent of vague responses with proteins or other biomolecules or metabolites [39]. For this purpose, the CNTs were initially surface-modified with PEG via a two-step activation of their carboxyl groups using EDC and NHS, and their subsequent reaction with the amino groups of PEG.

To conjugate all purified proteins onto the PEGylated CNTs, the recombinant proteins (hPTEN1, hPTEN2 or hPTEN3) were added into the solution of the PEGylated CNTs, following a second activation of the protein carboxyl groups and their reaction with the PEG amino groups, as described in Section 4.7. Any unconjugated protein was removed from the biofunctionalized CNTs by filtration and extensive washing.

The successful immobilization of the recombinant proteins was estimated by infrared spectroscopy (IR), zeta-potential measurements and thermogravimetric analysis (TGA), as previously described [57], and more information is given in Appendix A. In parallel to the physicochemical characterization of hPTEN1 conjugation to CNTs, SEM analysis (Figure 1) was performed to depict and verify the successful immobilization.

### 2.3. In Vitro Cell Viability

The biological effect of the biofunctionalized CNTs with PTEN and its variants were investigated by studying the cell viability of T-47D and PTEN-deficient ZR-75-1 cells, as well as in the non-PTEN deficient breast cancer cell line, MCF-7. As described in Section 4.12, T-47D, and ZR-75-1 cells were cultured for 24 h and, subsequently, various concentrations of CNTs, PEGylated CNTs, and biofunctionalized CNTs with all three proteins were added. Cell viability was estimated 48 and 96 h after the incubation.

As demonstrated in Table 1, the PEGylated CNTs displayed negligible cytotoxicity in T-47D cells after 48 h and 96 h incubation at all concentrations. In addition, the incubation of the T-47D cells for 48 h with PEGylated CNTs did not affect the number of the living cells comparing to non-treated control cells (Table 1). In contrast, incubation with the hPTEN1-functionalized CNTs did not only inhibit cell proliferation at low concentrations (20, and 50 μg/mL) but also reduced the cell viability by 50% at the concentration of 75 μg/mL (Table 1). However, the influence of the immobilized recombinant hPTEN1 was more effective after 96 h of incubation. More specifically, the treatment of T-47D cells with various concentrations (20, 50, and 75 μg/mL) triggered rapid reduction in the cell viability, up to 90%, as compared with the corresponding number of cells that were incubated with unconjugated nanotubes (Table 1). It is obvious that the optimal time of incubation was 96 h, whereas the inhibition of cell proliferation and the reduction of living cells already occurs at 48 h, at high concentrations. The above results indicate the ability of CNTs to transfer recombinant hPTEN1 to T-47D cells and maintain PTEN tumor suppressor activity.

In contrast, the biofunctionalized CNTs, with the two truncated fragments of PTEN (hPTEN2 and hPTEN3) assessed at concentrations of 20, 50 and 75 μg/mL, inhibited cell proliferation to a small extent only at the concentration 75 μg/mL, both at 48 and 96 h (Table 1). This result was expected for hPTEN3 since it consists of the C-terminal domain of PTEN that lacks the catalytic domain. For hPTEN2, which possesses the catalytic domain, our data revealed that the C-terminal domain is crucial for PTEN tumor suppressor activity.

T-47D cells showed the main morphological characteristics of epithelial cells, grown as a monolayer and considered as *PIK3CA*-mutant cell lines with elevated p-AKT expression (Appendix A). No morphological changes were observed after the 48 and 96 h incubation of cells with PEGylated CNTs (Appendix A, respectively). In contrast, the incubation of T-47D cells with 50 μg/mL CNTs-hPTEN1 provoked morphological changes after 48 h, which intensified after 96 h (Appendix A). Remarkably, after 96 h of treatment the cell morphology changed dramatically, as the cells shrunk and obtained cell membrane extensions and protrusions, and a fibroblast-like morphology (Appendix A, in the circle). This drastic morphological shift was accompanied by the identification of apoptotic cells exhibiting distinct features, such as the typical compact and rounded shape. Additionally, the images captured cells in the process of detachment, further emphasizing the impact of CNTs-hPTEN1 on inducing morphological changes associated with apoptosis. These findings underscore the potential of CNTs-hPTEN1 in influencing cellular processes and warrant further investigation into the underlying mechanisms driving these observed effects.

To determine the biological effect of the exogenous biofunctionalized CNTs-hPTEN1, while excluding the endogenous PTEN, we utilized the PTEN-deficient ZR-75-1 breast cancer cell line. As previously described, only the effect of CNTs-hPTEN1 was determined on ZR-75-1 cell viability. CNTs-hPTEN2 and CNTs-hPTEN3 were omitted since they were not active enough against T-47D cells in comparison with CNTs-hPTEN1. It was found that the cytotoxicity was not significant after 96 h treatment with 50 and 100 μg/mL of PEGylated CNTs (Table 2), while at higher concentrations (150, 200, and 300 μg/mL) a dramatic reduction of ZR-75-1 cell viability was observed, thus suggesting that PEGylated CNTs are cytotoxic at these concentrations. When 50 μg/mL of CNTs-hPTEN1 were added to ZR-75-1 cells, cell proliferation was inhibited 96 h later. A crucial reduction of cell growth was demonstrated with 100 μg/mL of CNTs-hPTEN1 at the same time, pointing out the significant tumor suppressor activity of PTEN.

ZR-75-1 are metastatic, epithelial, estrogen (ER)—and progesterone receptor (PR)—positive cells that grow primarily as a monolayer (Appendix A). Following the incubation of cells with PEGylated CNTs after 72 h, no significant morphological alterations were recorded. On the contrary, the incubation of ZR-75-1 cells with 20 μg/mL CNTs-hPTEN1 triggered morphological changes in the cells after 48 h, which was even more noteworthy in a time- and dose-dependent way. Specifically, apoptotic characteristics were observed, such as cell shrinkage and cell membrane protrusions, as recorded in T-47D cells, while a confluence reduction was discernible.

To enable a more comprehensive understanding of the generalizability of our experimental outcomes, independent of p-AKT expression, a third breast cancer cell line, MCF-7, was chosen. This experimental model is characterized by differentiated mammary epithelium, estrogen (ER)—and progesterone receptor (PR)—positive, *PIK3CA*-mutant cells with low p-AKT expression. As expected, the PEGylated CNTs did not affect the viability and the morphology of MCF-7 cells after 72 h even in the highest concentration of 100 μg/mL, while the treatment of CNTs-hPTEN1 exerts a prominent effect in the viability and the morphology at lower concentrations. In particular, alterations are observed even from 20 μg/mL, while the most potent concentration, in which there is no substantial effect from the PEGylated CNTs, is 100 μg/mL, as depicted in Appendix A.

In parallel to our optical microscope results for MCF-7 cells, a tendency similar to that of ZR-75-1 cells was also recorded. Specifically, the cytotoxicity was not important with 50 and 100 μg/mL of PEGylated CNTs after 96 h treatment (Table 2), while at higher concentrations a dramatic decrease of MCF-7 cell viability was recorded, thus highlighting that PEGylated CNTs are cytotoxic at these concentrations, as observed in ZR-75-1 cells. While MCF-7 cells were incubated with 50 μg/mL of CNTs-hPTEN1, cell proliferation was inhibited 96 h after treatment, as expected based on the Trypan blue exclusion test. A significant reduction of cell growth was displayed with 100 μg/mL of CNTs-hPTEN1 at the same time, indicating the significant tumor suppressor activity of PTEN.

### 2.4. Evaluation of Relative PTEN Protein Levels in Cell Lysates

Subsequent to the incubation of ZR-75-1 and MCF-7 cells with different concentrations of CNTs-PEG or CNTs-hPTEN1, cells were harvested at various time points (48, 72, and 96 h post incubation), and the cells were lysed with RIPA buffer. Afterwards, lysates were subjected to Western blot analysis after separation with SDS-PAGE 10% *w*/*v* for PTEN and phosphorylated PTEN in Ser_380_, since the phosphorylated PTEN in Ser_380_ leads to a loss of phosphatase activity and tumor suppressor function diminishment. For this reason, we evaluated PTEN protein activity by estimating the PTEN/P-PTEN (Ser_380_) ratio, the increase of which is correlated with a greater proportion of active and dephosphorylated PTEN and tumor suppression enhancement. The relative PTEN protein levels in ZR-75-1 and MCF-7 cells were assessed by estimating PTEN or P-PTEN in Ser_380_ band intensity, calculating the desirable ratio, and given below (Figure 2 and Figure 3). Cells not subjected to any treatment serve as the control cells, where the PTEN/P-PTEN band intensity was considered as 1, and all the treatments, CNTs-PEG or CNTs-hPTEN1, were normalized based on the control. Statistical analysis was performed for ZR-75-1 and MCF-7 CNTs-hPTEN1-treated cell extracts. Based on our cytotoxicity studies, CNTs-PEG did not influence cell population and morphology and, as a result, ZR-75-1 and MCF-7 CNTs-PEG-treated cell extracts did not undergo statistical analysis.

The PTEN-deficient cell line, ZR-75-1, have a basal, slight expression of PTEN in our experimental conditions. When the cells treated with different concentrations of CNTs-PEG, a decrease in the PTEN/P-PTEN ratio was recorded. Specifically, the decrease was greater with increasing concentration. In contrast, an increase in the PTEN/P-PTEN ratio was observed in ZR-75-1 cells treated with CNTs-hPTEN1 with increasing concentrations, thus pointing out that the higher CNTs-hPTEN1 concentration, the higher the relative PTEN activity in cells. As a result, a dose-dependent correlation was found, where CNTs-hPTEN1 with the concentration of 100 μg/mL delivered the highest amount of active PTEN in the cells. In parallel, a time-dependent correlation was not recorded, since the relative PTEN activity was decreased in the 75 μg/mL and 100 μg/mL at 72 and 96 h post CNTs-hPTEN1 treatments, respectively.

The results obtained from the experiments in MCF-7 cells exhibited many differences when juxtaposed with those noticed in ZR-75-1 cells. At first, the relative PTEN activity appeared to remain stable, showing no decrease or fluctuation between the control and CNTs-PEG in nearly all concentrations and time points. While a decline was observed at 100 μg/mL after 72 and 96 h post-incubation, the difference was far less pronounced compared to ZR-75-1. In the case of CNTs-hPTEN1 treatments, a dose- and time-dependent correlation was observed, with the concentration of 100 μg/mL identified as optimal for delivering the highest amount of active PTEN. This efficacy peak was reached 96 h post-treatment. The disparities in relative PTEN activity observed between ZR-75-1 and MCF-7 cells indicate that efficient PTEN delivery depends not only on the endogenous PTEN status (e.g., expression levels, post-translational modifications, regulation by other enzymes or miRNAs) but also on the cellular context. This dependence is attributed to the interplay of various signaling pathways.

The efficient PTEN protein delivery either in ZR-75-1 and MCF-7 cells as compared to the control cells, in conjunction with its tumor-suppressive function and morphological changes with apoptotic characteristics, represent the catalyst for the further study of the phenomenon and confirmation of apoptosis at the molecular level.

### 2.5. Investigation of CNTs-PEG and CNTs-hPTEN1 Potential in Cell Proliferation and Apoptosis

To further validate the role of CNTs-PEG and CNTs-hPTEN1 in cell proliferation and apoptosis, RT-PCR reactions were performed, with whole cytoplasmic RNA serving as the template. To examine the extent of promoting or repressing the proliferation capacity of ZR-75-1 and MCF-7 cells, a positive and a negative regulator of cell cycle were chosen. The mRNA of *PCNA* served as the positive regulator, while the mRNA of *CDKN1A (p21)* represented the negative regulator. To investigate the apoptosis induction in the two cell lines, the mRNA of a pro-apoptotic gene, *Bax*, and the mRNA of an anti-apoptotic gene, *Bcl-2*, were selected. The amplified products, in particular, were resolved by electrophoresis through 2% *w*/*v* agarose gel and ethidium bromide staining, and the appropriate bands were semi-quantified relative to the control non-treated cell lysates.

Regarding the cell proliferation, we found an expression decrease in *PCNA* mRNA in CNTs-hPTEN1-treated ZR-75-1 and MCF-7 cells as compared to CNTs-PEG-treated and control cells (Figure 4). Both for ZR-75-1 and MCF-7 cells, the greatest decrease was found when 100 μg/mL CNTs-hPTEN1 were utilized, which is in an absolute agreement with the Trypan blue assay (Table 2). On the other hand, CNTs-PEG did not influence the expression levels of *PCNA*, and its levels did not fluctuate. Moreover, the *CDKN1A (p21)* seemed to be inversely proportional regarding *PCNA* (Figure 5), an expected phenomenon because of their role and interplay in cell cycle. Our results indicated an increased expression in ZR-75-1 and MCF-7 CNTs-hPTEN1-treated cells as compared with the CNTs-PEG-treated and control cells. Also, it was observed that the highest increase was recorded in 100 μg/mL CNTs-hPTEN1, which is more solid proof for the reduction of ZR-75-1 and MCF-7 cell proliferation capacity, owing to the successful delivery of the recombinant hPTEN1 to CNTs.

To inspect the apoptotic potential of CNTs-hPTEN1 and CNTs-PEG in ZR-75-1 and MCF-7 cells, the *BAX/Bcl-2* mRNA expression ratio was calculated. In particular, an elevated value of this ratio corresponds to a higher expression of *Bax* and promotes cell death by mitochondrial outer membrane permeabilization, leading to the release of pro-apoptotic factors, including cytochrome c, into the cytoplasm. On the other hand, *Bcl-2* opposes the action of pro-apoptotic proteins, such as BAX, and assists in maintaining mitochondrial integrity. Overall, the *Bax/Bcl-2* expression ratio is often used as an indicator of the susceptibility of cells to undergo apoptosis. A higher *Bax/Bcl-2* ratio generally demonstrates a pro-apoptotic environment, favoring cell death, while a lower ratio reflects an anti-apoptotic environment, promoting cell survival.

Our results underlined the apoptotic potential of CNTs-hPTEN1, even in lower concentrations such as 20 μg/mL, where an increase in Bax/Bcl-2 ratio was first recorded in comparison with the control and CNTs-PEG lysates. Moreover, the apoptotic effect is stronger with a higher CNTs-hPTEN1 concentration, and at 100 μg/mL, a 3-times increase in the above-mentioned ratio was measured, as depicted in Figure 6. In parallel with ZR-75-1, MCF-7 cells displayed a similar trend. Notably, susceptibility to apoptosis was observed at concentrations of 10 μg/mL in CNTs-hPTEN1, compared to the control and CNTs-PEG counterparts. A dose-dependent correlation was evident, as increasing concentrations of CNTs-hPTEN1 led to an elevation in the Bax/Bcl-2 ratio. At a concentration of 100 μg/mL, a four-fold increase in the ratio was noted, suggesting that MCF-7 cells were more predisposed to undergo apoptosis compared to PTEN-deficient ZR-75-1 cells. Conversely, the Bax/Bcl-2 ratio remained unchanged in the ZR-75-1 and MCF-7 cells treated with CNTs-PEG, indicating that apoptosis could be attributed to recombinant hPTEN1. These findings are consistent with both the Trypan blue exclusion test and optical microscopy cell photos. (Table 2 and Appendix A).

### 2.6. Protein Expression Evaluation

To elucidate the role of CNTs-hPTEN1 in apoptosis induction, we performed a PathScan analysis using the Apoptosis Multi-Target Sandwich ELISA Kit to determine the levels of PTEN-dependent cell death in ZR-75-1 breast cancer cells. This kit detects the endogenous levels of key signaling proteins in the pathways controlling survival and apoptosis, such as total p53, phospho-p53 (Ser15), total Bad, phospho-Bad (Ser112), cleaved caspase-3 (Asp175) and cleaved PARP (Asp214). The activation of these proteins can be observed over time in response to CNTs-hPTEN1.

As depicted in Figure 7, the apoptosis results are presented as the absorbance at 450 nm of the HRP-linked secondary antibody per ZR-75-1 population, incubated with 100 μg/mL of CNTs-PEG, 50 and 100 μg/mL of CNTs-hPTEN1 and, finally, cells with no treatment. Cells were incubated with the above-mentioned CNTs for 96 h. As shown in Figure 7, a small increase was observed only for cleaved caspase-3 and cleaved PARP after the treatment of ZR-75-1 cells with 100 μg/mL of CNTs-PEG. On the contrary, treatment with 50 and, especially, 100 μg/mL of biofunctionalized CNTs-hPTEN1 in ZR-75-1 resulted in a significant increase of all protein levels tested, indicating that CNTs-hPTEN1 induced the apoptosis in ZR-75-1 cells and this effect was exclusively due to the immobilized hPTEN1 onto CNTs.

To evaluate the apoptotic effects of CNTs-PEG and CNTs-hPTEN1 at the protein level, MCF-7 cells were lysed with RIPA buffer. Subsequently, the lysates underwent Western blot analysis for p53, phospho-p53 (Ser15), Bad, phospho-Bad (Ser112), cleaved caspase-3 (Asp175), and cleaved PARP (Asp214) following electrophoretic separation. Consistent results were observed in MCF-7 cells, wherein an increased expression of phospho-p53 (Ser15), p53, cleaved PARP (Asp214), and phospho-Bad (Ser112) was detected only in CNTs-hPTEN1-treated cells. Specifically, the expression of these proteins was higher with increased concentrations of CNTs-hPTEN1, indicating apoptosis induction was detected even at 50 μg/mL but reached its peak at a final concentration of 100 μg/mL, as illustrated in Figure 8. These findings align with the results from optical microscopy, the Trypan blue exclusion test, and RT-PCR analysis. Conversely, in comparison with ZR-75-1 cells, no expression of Bad or cleaved caspase-3 was found in either control or CNTs-PEG/CNTs-hPTEN1-treated cells, suggesting caspase-3 independent apoptotic pathways (representative blots are provided in Appendix A).

## 3. Discussion

The main target of our study was the inhibition of cancer cell survival by the immobilized recombinant human PTEN onto CNTs, in its unphosphorylated and active form. As mentioned in the introduction, PTEN is located mainly in the cytoplasm and secondarily in the cell nucleus. When growth and survival signals activate the PI3’K-AKT-mTOR axis, PTEN binds and dephosphorylates PIP3 phospholipids, which are the main activation molecules of this pathway, to PIP2, and results eventually in the inhibition of the survival and proliferation of cancer cells. However, a combination of certain conditions is required for PTEN to bind the cytoplasmic membrane, such as the conversion of the stable phosphorylated form to a non-phosphorylated active form, which is, however, vulnerable to degradation [1,8]. Finally, it is known that PTEN-associated cancers cause the revelation of a C-terminus truncated PTEN (351–403 aa residues) [58,59], leading to a much lower expression when compared with the wild-type protein, despite its accelerated degradation [60].

To determine the effect of PTEN on breast cancer cell lines, we cloned, overexpressed, and purified its full-length form, hPTEN1, and its truncated fragments hPTEN2 and hPTEN3 in *E. coli* cells. Then, to stabilize the protein’s formation and to enable its import into cancer cells, the different forms of the protein (hPTEN1 or hPTEN2 or hPTEN3) were immobilized on the surface of multiwalled carbon nanotubes. This delivery system was selected owing to its large surface, small size, and invasive capacity in lipid membranes. These CNTs were previously surface decorated with polyethylene glycol to avoid any potential cytotoxicity and increase their hydrophilicity.

T-47D, MCF-7, and PTEN-deficient ZR-75-1 breast cancer cell lines were used in our study. The PEGylated nanotubes (CNTs-PEG) showed no noticeable toxicity in these three breast cancer cell lines, especially at concentrations below 150 μg/mL. Most importantly, the significant inhibition of cell growth was observed at a concentration of 50 and 100 μg/mL of CNTs-hPTEN1. However, the two truncated forms of the protein, hPTEN2 and hPTEN3, displayed a lower ability to reduce the number of cells as compared to hPTEN1. Specifically, hPTEN2 harbors the catalytic domain, while the CBR3 loop is missing, thus affecting its ability to bind to the cell membrane and dephosphorylate membrane lipids. The limited potential of hPTEN2 to inhibit cell growth could be attributed to its reduced ability to bind to cellular membrane or to protein degradation. It has been reported that PTEN minus the C-terminus domain is susceptible to rapid degradation [16,33]. Meanwhile, TI and CBR3 (calcium-binding region) loops, encoded by residues 160–171 and 260–269, respectively, orient the active site in the appropriate position for the dephosphorylation of the membrane lipids [61,62].

In parallel to hPTEN2, hPTEN3 was expected to be inactive since it lacks the catalytic site. However, we recorded a decreased apoptotic potential compared with hPTEN1, a phenomenon that could be explained from Ahmed’s work [33]. Particularly, they suggest that the exosome-mediated delivery of the PTEN C-terminus domain could mediate the stabilization of intrinsic PTEN and, thus, promote anti-proliferative and anti-tumorigenic responses to different cancer cell lines [33]. Overall, our results indicate the necessity of all PTEN domains, regions, loops, or residues to exert their maximal tumor-suppressive activity.

In most cancers, defects in apoptotic stimuli and pathways are associated with cancer hallmarks [63]. Therefore, it became clear that cellular resistance to the apoptotic signals contribute to oncogenesis, tumor progression and treatment resistance. The impact of these processes on the cell population can be evaluated by observing alterations in various crucial signaling components. Such a tumor-suppressor gene is p53, which exerts an important role in the maintenance of genomic integrity, angiogenesis inhibition and apoptosis induction depending on the extent of cell damage. In cancer cells, the constant degradation of p53 leads to low protein levels, which agrees with our results after no treatment of both cell lines. PTEN governs p53 state and activity at the protein level [64], while it is also involved in G protein-coupled receptor (GPCR) transduction during chemoattractance [65]. More specifically, PTEN inhibits the mouse double minute 2 homolog (MDM2)—p53 protein binding (to the N-terminal trans-activation domain of p53), thus preventing p53 from being active and enhancing its cytoplasmic sequestration [66]. As a result, PTEN protects the tumor-suppressor protein p53 via the PI3K/AKT pathway as well as via the direct binding of p53 and, consequently, blocking anti-apoptotic processes [67]. This p53 expression level elevation was shown in this study in ZR-75-1 and MCF-7 cells after treatment with hPTEN1 biofunctionalized CNTs at a final concentration of 100 μg/mL, proving once again the efficient cellular delivery of the active protein form.

Another critical pro-apoptotic protein, caspase-3, is activated by endoproteolytic cleavage at Asp175. Caspase-3 exerts its activity through the cleavage of multiple cellular targets, such as the DNA repair and apoptosis nuclear enzyme poly(ADP-ribose) polymerase (PARP) (cleavage at Asp214) [68,69]. We found elevated expression levels of phospho-p53 (Ser15), p53, cleaved-PARP (Asp214) and phospho-Bad (Ser112) in both CNTs-hPTEN1-treated ZR-75-1 and MCF-7 cells in a dose-dependent manner. The augmented expression of the above-mentioned proteins was not detected in both ZR-75-1 and MCF-7 CNTs-PEG-treated and control cells, thus underlining that the apoptotic effect could be attributed to the successful hPTEN1 delivery. Although significant increased expression levels of cleaved caspase-3 and Bad were recorded in ZR-75-1 cell lysate, the same effect was not noticed in MCF-7 cells; therefore, the PARP cleavage at Asp214 occurs independently of caspase-3 activation. Simultaneously, our results suggest that the effect of CNTs-hPTEN1 on caspase-3 activation is cell-type specific and cell-context dependent, since the induction of caspase-3 enzymatic activity was only observed in ZR-75-1 cells but not in CNTs-hPTEN1-treated MCF-7 cells. Our data support the notion that different apoptotic pathways may be activated and utilized by the same apoptotic stimulus in various cells.

Our findings align with those of other research teams in the field. While no other group has specifically worked with PTEN-biofunctionalized CNTs, numerous studies have explored the effects of the PTEN gene or mRNA in various cell lines and animal models. These studies consistently highlight the activation of apoptotic pathways and the suppression of cell proliferation through PTEN gene or mRNA delivery, which we have also observed. PTEN protein therapeutics appear to induce apoptosis at rates comparable to those achieved with nucleic acid-based therapeutics.

Cancer cell survival requires the inhibition of pro-apoptotic factor expression, as well as the expression enhancement of anti-apoptotic factors. It is well known that the PI3K/AKT pathway, activated by many survival factors, leads to the activation of AKT protein targets via phosphorylation. AKT inhibits the pro-apoptotic Bcl-2 family members, such as the Bcl-2-associated death promoter (Bad) protein, which protects the integrity of mitochondria, preventing cytochrome c release and the subsequent activation of caspase-9 [70]. Dephosphorylated Bad forms a heterodimer with Bcl-2 and Bcl-xL, thus inactivating them and allowing for Bax/Bak-triggered apoptosis [71]. This is consistent with the increased expression levels of Bad after treatment with 100 μg/mL CNTs-hPTEN1 in ZR-75-1 cells. However, the reduction of the phosphorylated Bad, which is known to promote cancer cell survival, was not observed in our results. In parallel, we found zero protein expression for Bad and cleaved-caspase 3 in any other group in MCF-7 cells, thus meaning that the apoptosis is induced independently of the activation of that proteins.

## 4. Materials and Methods

### 4.1. Materials

Restriction endonucleases, T4 DNA ligase and Vent™ polymerase were purchased from New England BioLabs, INC, Hertfordshire, UK. Molecular weight DNA and protein markers were obtained from Nippon Genetics (Nippon Genetics Europe GmbH, Düren, Germany). PCR primers for the cloning of hPTEN1, hPTEN2 and hPTEN3 were obtained from MWG, Germany, while primers for assessing the gene expression levels of different effectors were manufactured by Eurofin Genomics GmbH, Ebersberg Germany and are tabulated in Table 2. An expression vector containing the human PTEN open reading frame (pORF9-hPTEN v21) was purchased from InvivoGen, San Diego, CA, USA, and vectors pGEM-T Easy and pET-29c(+) were obtained from Promega (Promega Corporation, Madison, WI, USA) and Novagen (Novagen, Madison, WI, USA), respectively. All other chemicals were obtained from Sigma-Aldrich Chemie GmbH, Steinheim, Germany.

### 4.2. Bacterial Growth Conditions

*E. coli* were grown at 37 °C in Luria–Bertani (LB) medium [72] under vigorous shaking. Ampicillin and kanamycin were added in the culture media, when necessary, at final concentrations of 100 μg/mL and 50 μg/mL, respectively. The induction of recombinant hPTEN1, hPTEN2 and hPTEN3 expression in *E. coli* BL21[DE3] (Molecular Cloning Laboratories (MCLAB), San Francisco, CA, USA) carrying the expression plasmids pET29c-hpten1, pET29c-hpten2 and pET29c-hpten3, was achieved by IPTG addition (1 mM) to the cultures, when the absorbance at 600 nm was 0.6 (Table 3).

### 4.3. Bacterial Strains and Plasmids

The bacterial strains and plasmids used in this study are listed in Table 1. For the construction of the expression vector of recombinant hPTEN1, or its two truncated fragments (1–185 or 186–403 aa for hPTEN2 and hPTEN3, respectively) fused to a C-terminal His_6_-tag, the corresponding DNA sequences were amplified using pORF9-hPTEN vector as a template. The primer pairs used consisted of the oligonucleotides: *hpten1*: 5′-CAT ATG ACA GCC ATC ATC AAA GAG-3′ and 5′-CTC GAG GAC TTT TGT AAT TTG TGT A-3′, *hpten2*: 5′-CAT ATG ACA GCC ATC ATC AAA GA-3′ and 5′-CTC GAG TCC CTT TTT GTC TCT GG-3′, *hpten3*: 5′-CAT ATG CCC AGT CAG AGG CGC-3′ and 5′-CTC GAG GAC TTT TGT AAT TTG TGT A-3′. The 1221-bp, 504-bp and 719-bp PCR products, were TA-cloned into pGEM-T Easy vector (Promega) to yield pGEM-*hpten1*, pGEM-*hpten2* and pGEM-*hpten3* plasmids, respectively. All inserts were retrieved from pGEM-*hpten1*, pGEM-*hpten2* and pGEM-*hpten3* by *NdeI*/*XhoI* double digestion and, consequently, fragments were cloned into pET-29c expression vector (Novagen) to yield pET29c-*hpten1*, pET29c-*hpten2* and pET29c-*hpten3*, respectively.

### 4.4. Purification of Recombinant hPTEN Protein Forms

Transformed *E. coli* BL21[DE3] cells carrying pET29c-*hpten1*, pET29c-*hpten2* and pET29c-*hpten3* were grown at 37 °C in Luria–Bertani broth containing 50 μg/mL kanamycin and induced with 1 mM IPTG, as described above. Following harvest, cells were lysed and all proteins were purified with affinity chromatography using Protino Ni-TED resin columns (Macherey-Nagel), as described in [73]. During isolation, samples from the soluble fraction of cells (cytoplasmic fraction), the lipid-soluble fraction (membrane fraction), and the inclusion bodies were analyzed by SDS-PAGE and silver nitrate staining. The His6-tagged proteins hPTEN1, hPTEN2 and hPTEN3 were displayed in the expected range at the fraction of inclusion bodies. Therefore, further purification of the three proteins under denaturing conditions was achieved following the solubilization of inclusion body proteins by an 8 M urea solution in lysis buffer (50 mM NaH_2_PO_4_, 300 mM NaCl, 8 M Urea). The solubilized proteins were purified by Protino Ni-TED Resin columns (Macherey-Nagel). Samples from the inclusion bodies and the three elution fractions (50 mM, 50 mM and 250 mM imidazole) were analyzed by SDS-PAGE and silver nitrate staining. Protein refolding was achieved by the gradual removal of urea and dialysis in storage buffer (25 mM Tris–HCl pH 7.9, 150 mM NaCl, 5mM MgCl_2_, 5 mM β-mercaptoethanol, 10% (*v*/*v*) glycerol) containing 0.1% (*v*/*v*) Tween-20.

### 4.5. Electrophoresis and Western Blotting

SDS-polyacrylamide gel electrophoresis (SDS-PAGE) was performed using 10% or 12% (*w*/*v*) polyacrylamide gels containing 0.1% (*w*/*v*) SDS, as described by Laemmli et al. [74]. Protein concentrations were determined by the Bradford method [75], using bovine serum albumin as the reference standard. Proteins were stained with either Coomassie Brilliant Blue R250 [75,76] or silver nitrate [77], and electrotransferred to nitrocellulose membranes (Macherey-Nagel, Düren, Germany) following the method of Towbin et al. [78]. The primary antibodies used are illustrated below: polyclonal rabbit His-tag, monoclonal rabbit PTEN, monoclonal rabbit phospho-PTEN (Ser380), rabbit monoclonal Pan-Actin, mouse monoclonal p53, rabbit polyclonal phospho-p53 (Ser15), rabbit polyclonal Bad, rabbit polyclonal phospho-Bad (Ser112), rabbit polyclonal Caspase-3, and rabbit monoclonal PARP. All were obtained from Cell Signaling Technology, except for the mouse monoclonal p53, which was purchased from Santa Cruz Biotechnology. The AP-linked secondary antibodies, anti-rabbit and anti-mouse, were also purchased from Cell Signaling Technology. BCIP (Biotium, Fremont, CA, USA) and NBT (Sigma-Aldrich Chemie GmbH, Steinheim, Germany) were used for colorimetric detection of AP activity.

### 4.6. Decoration of CNTs with bis(3-aminopropyl)Polyethylene Glycol

As was previously described [57], oxidized CNTs (9.5 nm diameter × 1.5 nm length, Sigma-Aldrich Chemie GmbH, Steinheim, Germany) were firstly subjected to sonication, the carboxyl groups were activated by EDC/NHS, and bis(3-aminopropyl)PEG (MW 1500) was added. Finally, the CNTs were filtered by using a 0.45 μm polycarbonate filters (Millipore, Burlington, MA, USA) and washed with water to remove excess reagents, followed by freeze-drying to get purified PEG-CNTs.

### 4.7. Protein Conjugation

For the protein conjugation of the surface of PEGylated CNTs, hPTEN1, hPTEN2 and hPTEN3 were added to the mixture at a concentration of 1 mg/mL each to the CNTs-PEG dispersion (2mg), and the same procedure was followed as previously described [57]

### 4.8. Fourier Transform Infrared (FTIR) Spectroscopy

The chemical structure of the immobilized proteins onto CNTs was studied by recording their FTIR spectra (Perkin-Elmer, Spectrum One, Akron, OH, USA), according to [57]. The samples were made into pellets with potassium bromide, and a total of 64 spectra were averaged to reduce noise. All spectra were recorded at a resolution of 4 cm^−1^ and the recorded wavenumber range was 800–4000 cm^−1^. The commercially available software Spectrum v5.0.1 (Perkin Elmer LLC 1500F2429) (Perkin-Elmer, Spectrum One, Akron, OH, USA) was used to process the spectral data.

### 4.9. Thermogravimetric Analysis (TGA)

TGA analysis was performed on a Pyris 1 TGA (Perkin-Elmer, Akron, OH, USA) thermal analyzer. Each sample was accurately weighted (10 mg) and heated from ambient temperature to 700 °C at a heating rate of 20 °C min^−1^ under constant nitrogen flow, according to [57]. Based on our cytotoxicity results on T-47D cells using the Trypan blue method, and given that CNTs-hPTEN1 was selected for further studies in cell lines, the TGA analysis was performed exclusively on CNTs-hPTEN1.

### 4.10. Zeta Potential

Zeta potential calculations were conducted by a dynamic light scattering (DLS) analyzer (Zetasizer Nano, Malvern Instruments, Nano ZS, ZEN 3600, Malvern, UK). The ζ-potential values were measured after resuspending the PEGylated CNTs (CNTs-PEG) and the hPTEN1 biofunctionalized CNTs (CNTS-hPTEN1) in an aqueous solution of PBS. Experiments were carried out in triplicates for each sample, and the ζ-potential value represents the mean average. Zeta potential measurements were carried out at room temperature (25 °C) at a back scatter angle of 175° on samples diluted to 1/100 of their original concentrations, which were 9.56 mg/mL for CNTs-PEG and 5.4 mg/mL for CNTs-hPTEN1.

### 4.11. Surface Morphology Imaging by Scanning Electron Microscopy (SEM)

An amount of 2 mL of CNTs-PEG and CNTs-hPTEN1 stock solutions were subjected to lyophilization, attached on the specimen stubs, and coated by carbon using a JEE-4X vacuum evaporator to achieve electrical conductivity for observation by SEM (JEOL J.S.M. 840A, Tokyo, Japan), equipped with an AZTEC ENERGY ADVANCED X-act EDS (Oxford, UK) analyzer, operating at 5.0. and 10.0 kV at the Electron Microscopy and Structural Characterization of Materials Laboratory of the Department of Physics at Aristotle University of Thessaloniki. Micrographs were taken at a magnification of ×5000 and ×10,000.

### 4.12. Cell Cultures, Cell Lysis and In Vitro Cell Viability

Human breast carcinoma T-47D (HTB-133), ZR-75-1 (CRL-1500) and MCF-7 (HTB-22) cell lines were obtained from ATCC (American Type Culture Collection, Manassas, VA, USA). Τ-47D and ZR-75-1 breast cancer cells were cultured in Roswell Park Memorial Institute medium (RPMI)-1640 medium supplemented with 10% (*v*/*v*) fetal bovine serum (FBS) (Thermo Fischer, Gibco, Waltham, MA, USA), 1% (*w*/*v*) penicillin and streptomycin (Thermo Fischer, Gibco, Waltham, MA, USA). MCF-7 breast cancer cells were cultivated in DMEM medium containing 4.5 g/L glucose with L-glutamine and sodium pyruvate (Fischer, Gibco, Waltham, MA, USA), 10% (*v*/*v*) fetal bovine serum, and 1% (*v*/*v*) penicillin/streptomycin. The cells were maintained at 37 °C in a humidified 5% (*v*/*v*) CO_2_.

Control cells, cells incubated with different concentrations of CNTs-PEG or CNTs-hPTEN1 at different time points, grown at a confluency of 70–80% in 6-well plates, were washed twice in ice-cold phosphate-buffered saline (PBS), lysed using an appropriate volume of RIPA Lysis Buffer (50 mM Tris-HCl pH 7.5, 150 mM NaCl, 1% (*v*/*v*) Triton X-100, and 1 mM PMSF) depending on cell population, and harvested from culture dishes on ice using a cell scraper. After being passed within a 25-gauge syringe needle eight to ten times, the whole extract was heated at 95 °C for 2 min. Whole-cell extracts were centrifuged for 10 min at 10,000× *g* at 4 in a microcentrifuge for efficient clarification, then separated at 50 µg of total protein lysates per well on 10% SDS-PAGE, and transferred to nitrocellulose membrane for Western blot analysis, as mentioned above. The total protein concentration was calculated using the Bradford Assay.

8 × 10^4^ cells were seeded out in 24-well dishes using RPMI with 10% *v/v* FBS. After 24 h maintenance, the medium was switched with fresh one in conjunction with different concentrations of the CNTs-hPTEN1. Cell viability was determined by the Trypan blue dye exclusion test. Cells were treated with a 0.5% (*w*/*v*) trypsin/EDTA solution and counted using a microscope Neubauer hemocytometer at the indicated time points.

### 4.13. Total RNA Isolation and Semi-Quantitative RT-PCR

Whole cytoplasmic RNA was extracted from non-treated and CNTs-PEG- or CNTs-hPTEN1-treated cells using the NucleoSpin RNA kit (Macherey-Nagel, Düren, Germany). The PrimeScript RT Reagent Kit-Perfect Real Time (Takara Bio Inc., Kusatsu, Shiga, Japan cDNA was utilized for cDNA production. Semi-quantitative PCR reactions were performed using the Q5^®^ High-Fidelity DNA Polymerase (New England BioLabs, INC). The PCR conditions were the following: an initial denaturation step at 95 °C for 20 s, followed by 35 cycles of amplification (denaturation at 98 for 10 s, annealing at 56–65 °C for 30 s and extension at 72 °C for 60 s), with a final extension at 72 °C for 2 min. All reactions were performed in triplicates. Amplified products were resolved by electrophoresis through 1.5 or 2% *w/v* agarose gel and ethidium bromide staining. The relative expression of mRNAs studied was estimated based on band intensity through appropriate results processing with ImageJ 1.53t software (https://imagej.net/software/imagej/ (accessed on 27 May 2024), and the data were normalized to GAPDH levels. The primers used for the RT-PCR analysis are listed in Table 4.

### 4.14. PathScan Analysis

To determine the effect of the hPTEN1-biofunctionalized CNTs (CNTs-hPTEN1) on ZR-75-1 and MCF-7 cancer cells, PathScan analysis was performed using the PathScan Apoptosis Multi-Target Sandwich ELISA Kit (Cell Signaling Technology, Cat. No. 7105), according to the manufacturer’s instructions. This kit detects endogenous levels of total p53, phospho-p53 (Ser15), total Bad, phospho-Bad (Ser112), cleaved caspase-3 (Asp175) and cleaved PARP (Asp214). Briefly, after 96 h of incubation with biofunctionalized carbon nanotubes at a concentration of 100 μg/mL, ZR-75-1 and MCF-7 cells were collected by centrifugation and 0.2 mL of ice-cold 1X Cell Lysis Buffer, plus 1 mM phenylmethylsulfonyl fluoride (PMSF), were added to each tube on ice for 5 min. Cells were sonicated for 3 min and centrifuged for 10 min at 4 °C. After the collection of whole-cell lysates, protein concentration was determined, and the proteins were analyzed at equal concentrations as follows: A total of 100 μL of sample diluent (supplied in each PathScan^®^ Apoptosis Multi-Target Sandwich ELISA Kit #7105) were added to a microcentrifuge tube with 100 μL of cell lysate and vortexed for a few seconds. Then, 100 μL of each diluted cell lysate were added to the appropriate microwell. Microwells were sealed with tape and incubated overnight at 4 °C. Afterwards, the plate content was discarded, the microwells were washed 4 times with 200 μL of 1X Wash Buffer, and incubated with 100 μL of each Detection Antibody for 1 h at 37 °C. The wash procedure was repeated as mentioned before. In total, 100 μL of HRP-linked secondary antibody were added to each well and the plate was incubated for 30 min at 37 °C. The wash procedure was repeated, 100 μL of TMB substrate were added to each well and the plate was incubated for 10 min at 37 °C. Finally, 100 μL of STOP Solution were added to each well and the plate was shaken gently for a few seconds. The spectrophotometric determination of results was obtained by reading the absorbance at 450 nm within 30 min after adding STOP Solution.

### 4.15. Statistical Analysis

PCR results are depicted as the mean ± SD of experiments. The relative expression of all target transcripts and proteins was calculated using ImageJ 1.53t software (https://imagej.net/software/imagej/ (accessed on 27 May 2024)) and is represented as the mean of three independent experiments in triplicates. Student’s *t*-test was exploited for checking the differences between the distinct groups, with the statistical significance set at *p* = 0.05. The proper depiction for diagrams with the corresponding statistical analysis was prepared using GraphPad Prism (Version 8.0.1) (https://www.graphpad.com/ (accessed on 27 May 2024)).

## 5. Conclusions

In conclusion, this work investigates the promising realm of utilizing multiwalled CNTs as carriers for PTEN and its truncated fragments in breast cancer cells. The efficacy of multiwalled CNTs as a PTEN delivery system was revealed, as well as the necessity of using the full-length PTEN in inhibiting cancer growth and inducing apoptosis.

Moreover, the findings of this work underscore the critical role of the C2 domain in the C-terminal region for the full-length PTEN’s ability to inhibit cancer cell growth and induce apoptosis in T-47D, ZR-75-1 and MCF-7 cells. Intriguingly, the truncated PTEN fragments, despite possessing the active site, exhibited only slight effects, emphasizing the nuanced nature of PTEN’s functionality.

The successful immobilization of full-length PTEN onto PEGylated CNTs demonstrated robust efficacy in inhibiting cancer cell growth, altering the expression levels of key pro-apoptotic and anti-apoptotic transcripts and proteins. This intricate understanding of PTEN’s behavior and the potential of CNTs as delivery systems provide crucial insights for the future research and development of advanced strategies in the delivery of tumor suppressor proteins for cancer therapy.

## Figures and Tables

**Figure 1 molecules-29-02785-f001:**
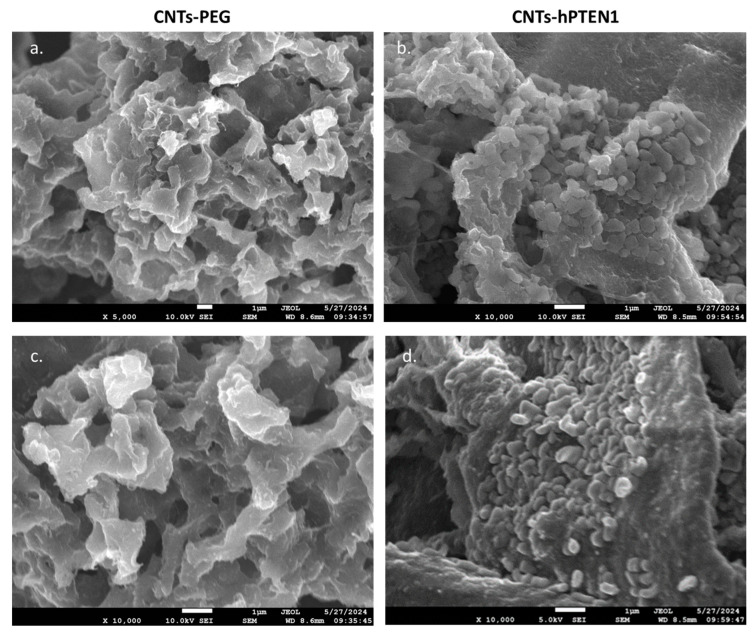
SEM analysis of CNTs-PEG at magnification 5000× (**a**) and 10,000× (**c**) and CNTs-hPTEN1 at magnification 10,000× (**b**,**d**).

**Figure 2 molecules-29-02785-f002:**
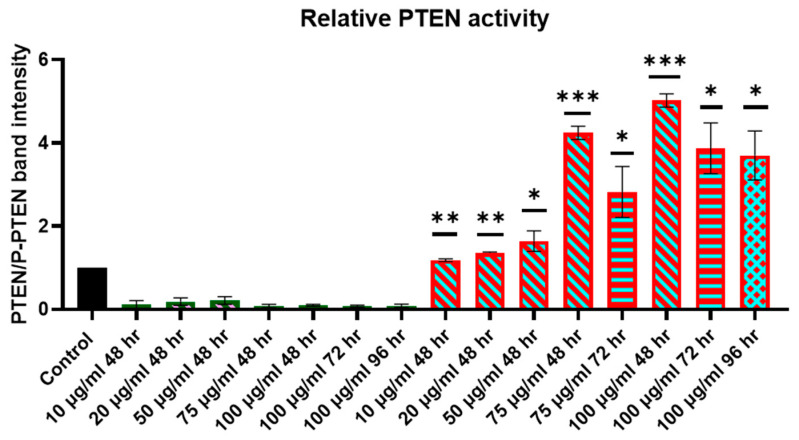
The relative PTEN activity of ZR-75-1 breast cancer cells incubated with CNTs-PEG and CNTs-hPTEN1, as measured from the PTEN/P-PTEN Western blot analysis band intensity. Statistical analysis was performed using GraphPad Prism software (Version 8.0.1) (GraphPad, San Diego, CA, USA) Statistical significance was determined using the *p*-value (*p* < 0.05 (represented by *), *p* < 0.01 (represented by **), *p* < 0.001 (represented by ***)).

**Figure 3 molecules-29-02785-f003:**
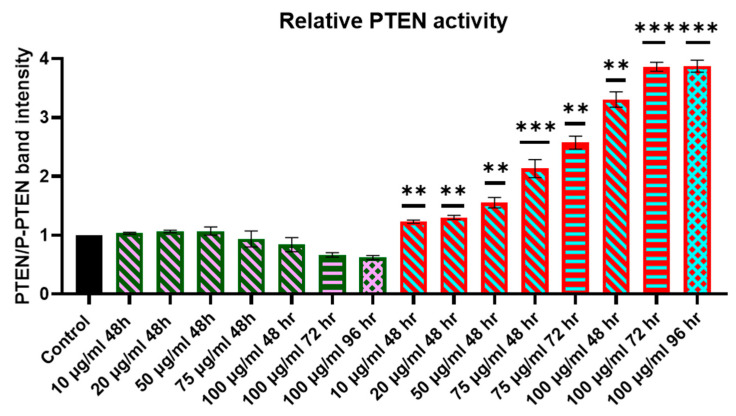
The relative PTEN activity of MCF-7 breast cancer cells incubated with CNTs-PEG and CNTs-hPTEN1, as measured from the PTEN/P-PTEN Western blot analysis band intensity. Statistical analysis was performed using GraphPad Prism software (Version 8.0.1) (GraphPad, San Diego, CA, USA) Statistical significance was determined using the *p*-value (*p* < 0.01 (represented by **), *p* < 0.001 (represented by ***)).

**Figure 4 molecules-29-02785-f004:**
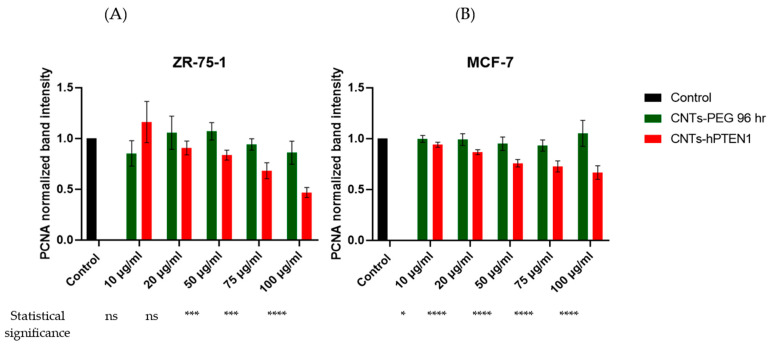
The *PCNA* mRNA normalized intensity ratio of ZR-75-1 (**A**) and MCF-7 (**B**) breast cancer cells treated with CNTs-PEG and CNTs-hPTEN1. Statistical analysis was performed using GraphPad Prism software (Version 8.0.1) (GraphPad, San Diego, CA, USA) by multiple *t* tests. Statistical significance was determined using the *p*-value (*p* < 0.05 (represented by *), *p* < 0.001 (represented by ***) and *p* < 0.0001 (represented by ****), whilst “ns” indicates no-significant differences.).

**Figure 5 molecules-29-02785-f005:**
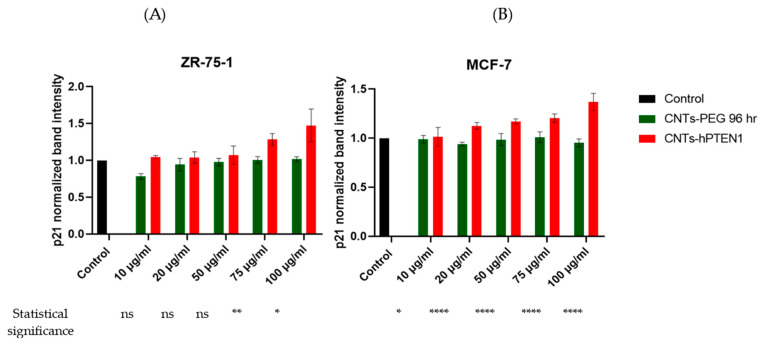
The *CDKN1A (p21)* mRNA normalized intensity ratio of ZR-75-1 (**A**) and MCF-7 (**B**) breast cancer cells treated with CNTs-PEG and CNTs-hPTEN1. Statistical analysis was performed using GraphPad Prism software (Version 8.0.1) (GraphPad, San Diego, CA, USA) by multiple *t* tests. Statistical significance was determined using the *p*-value (*p* < 0.05 (represented by *), *p* < 0.01 (represented by **), *p* < 0.0001 (represented by ****), whilst “ns” indicates no-significant differences).

**Figure 6 molecules-29-02785-f006:**
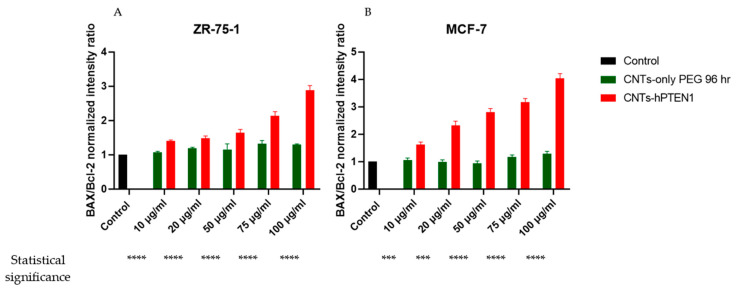
The *Bax/Bcl-2* mRNA normalized intensity ratio of ZR-75-1 (**A**) and MCF-7 (**B**) breast cancer cells treated with CNTs-PEG and CNTs-hPTEN1. Statistical analysis was performed using GraphPad Prism software (Version 8.0.1) (GraphPad, San Diego, CA, USA) by multiple *t* tests. Statistical significance was determined using the *p*-value (*p* < 0.001 (represented by ***) and *p* < 0.0001 (represented by ****)).

**Figure 7 molecules-29-02785-f007:**
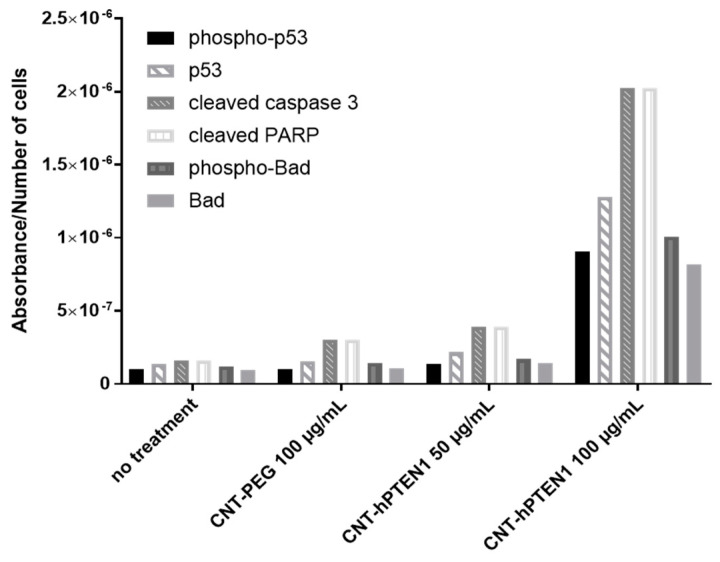
PathScan apoptosis analysis. The absorbance at 450 nm of HRP-linked secondary antibody per ZR-75-1 population was estimated for cells incubated with 100 μg/mL of CNTs-PEG, 50 and 100 μg/mL of biofunctionalized CNTs-hPTEN1 and after no treatment (control) for 96 h.

**Figure 8 molecules-29-02785-f008:**
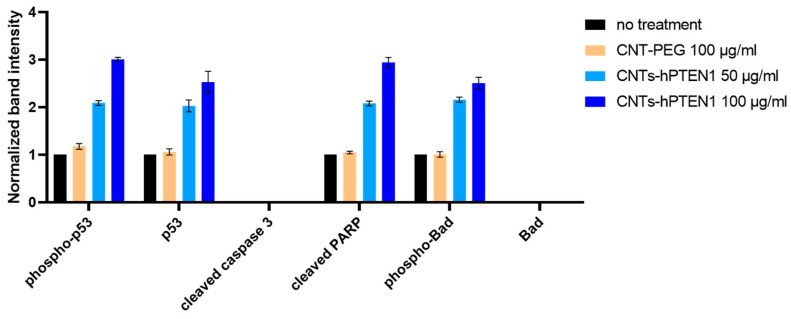
Western blot analysis of MCF-7 cell extracts treated for 96 h with 100 μg/mL of CNTs-PEG, 50 and 100 μg/mL of biofunctionalized CNTs-hPTEN1 and after no treatment (control).

**Table 1 molecules-29-02785-t001:** Inhibition of cell proliferation of T-47D cells after treatment with 20, 50 and 75 μg/mL PEGylated carbon nanotubes (CNT-PEG), biofunctionalized carbon nanotubes CNT-hPTEN1, CNT-hPTEN2 and CNT-hPTEN3 for 48 and 96 h.

	Concentration (μg/mL)		Mean Cell Population	SD	Survival Rate (%)	Statistical Significance
**48 h**	0	no treatment	145,666.6	32,024.4		
	20	CNTs-PEG	146,702.7	31,750.4	100.7	
		CNTs-hPTEN1	118,333.3	34,725.7	81.2	**
		CNTs-hPTEN2	141,666.8	35,910.02	97.2	ns
		CNTS-hPTEN3	143,333.2	34,975.4	98.3	ns
	50	CNTs-PEG	146,812.1	38,621.4	100.7	
		CNTs-hPTEN1	96,666.9	24,675.2	66.3	***
		CNTs-hPTEN2	136,315.5	22,317.8	93.5	ns
		CNTS-hPTEN3	154,343.5	27,642.6	105.9	ns
	75	CNTs-PEG	137,658.9	29,875.5	94.5	
		CNTs-hPTEN1	50,048.4	21,426.1	34.3	****
		CNTs-hPTEN2	97,467.1	19,924.6	66.9	***
		CNTS-hPTEN3	99,123.8	20,067.8	68.0	***
**96 h**	0	no treatment	142,334.3	29,475.2		
	20	CNTs-PEG	145,435.6	32,515.2	102.1	
		CNTs-hPTEN1	116,232.1	34,500	81.6	**
		CNTs-hPTEN2	136,475.1	25,605.2	95.8	ns
		CNTS-hPTEN3	143,654.7	28,326.8	100.9	ns
	50	CNTs-PEG	144,097.2	39,657.2	101.2	
		CNTs-hPTEN1	99,754.3	29,245.4	70.0	***
		CNTs-hPTEN2	138,225.4	25,675.2	97.1	ns
		CNTS-hPTEN3	145,091.1	27,450.1	101.9	ns
	75	CNTs-PEG	127,400.1	16,758.1	89.5	
		CNTs-hPTEN1	49,435.2	20,043.7	34.7	****
		CNTs-hPTEN2	99,655.1	17,575.2	70.0	***
		CNTS-hPTEN3	98,442.3	19,878.2	69.1	***

Statistical analysis was performed using GraphPad Prism software (Version 8.0.1) (GraphPad, San Diego, CA, USA) by multiple *t* tests, followed by Holm–Sidak method analysis. Values are presented as mean + SD, n = 10 per group. Statistical significance was determined using the *p*-value, *p* < 0.01 (represented by **), *p* < 0.001 (represented by ***) and *p* < 0.0001 (represented by ****), whilst “ns” indicates no-significant differences.

**Table 2 molecules-29-02785-t002:** Inhibition of cell proliferation after incubation of ZR-75-1 and MCF-7 cells with 0, 50, 100 and 150 μg/mL PEGylated carbon nanotubes (CNTs-PEG) and treatment with the biofunctionalized CNTs-hPTEN1 for 96 h.

	Concentration (μg/mL)		Μean Cell Population	SD	Survival Rate (%)	Statistical Significance
ZR-75-1	0	no treatment	769,230.0	62,142.2	100.0	
	50	CNTs-PEG	702,564.1	1,194,551.9	91.3	*
		CNTs-hPTEN1	523,076.9	127,454.5	68.0	
	100	CNTs-PEG	610,256.4	14,514.5	79.3	****
		CNTs-hPTEN1	82,051.3	16,964.1	10.7	
	150	CNTs-PEG	471,794.9	12,925.0	61.3	****
		CNTs-hPTEN1	30,769.2	14,943.2	4.0	
MCF-7	0	no treatment	807,692.3	30,265.1	100.0	
	50	CNTs-PEG	721,538.5	10,658.8	89.3	****
		CNTs-hPTEN1	352,307.7	74,181.9	77.7	
	100	CNTs-PEG	627,692.3	4615.4	51.4	****
		CNTs-hPTEN1	220,000.0	47,368.6	43.6	
	150	CNTs-PEG	415,384.6	36,633.5	27.2	****
		CNTs-hPTEN1	23,076.9	12,211.2	2.9	

Statistical analysis was performed using GraphPad Prism software (Version 8.0.1) (GraphPad, San Diego, CA, USA) by multiple *t* tests, followed by Holm–Sidak method analysis. Values are presented as mean + SD, n = 3 per group. Statistical significance was determined using the *p*-value (*p* < 0.05 (represented by *) and *p* < 0.0001 (represented by ****)).

**Table 3 molecules-29-02785-t003:** Characteristics of bacterial strains and plasmids.

*E. coli* Strains or Plasmids	Strain Genotype or Plasmid Information	Source
JM109	*F^−^, mcrA, Δ(mrr-hsdRMS-mcrBC), Ø80lacZΔM15, ΔlacX74, recA1, endA1*	InvivoGen
TOP10	*F^−^, mcrA, Δ(mrr-hsdRMS-mcrBC), φ80lacZΔM15, ΔlacX74, recA1, araD139, Δ(ara-leu) 7697, galU, galK, rpsL (Str^R^), endA1, nupG, λ-*	Invitrogen (Waltham, MA, USA)
BL21 [DE3]	*F^−^, ompT, gal, dcm, lon, hsdS_B_(r_B_^-^ m_B_^-^), λ(DE3 [lacI, lacUV5-T7 gene 1, ind1, sam7, nin5])*	Molecular Cloning Laboratories: MCLAB
pORF9-*hPTEN* v21	Expression vector containing the human PTEN open reading frame	InvivoGen
pGEM-T Easy	pGEM-T Easy vector carrying T-overhangs at the EcoRI site for TA cloning	Promega
pGEM-*hpten1*	PCR product of hpten ORF in vector pGEM-T Easy—Amp^r^	This study
pGEM-*hpten2*	PCR product of sequences encoding the N-terminal region (a.a. 1–185) of *hpten* cloned in vector pGEM-T Easy—Amp^r^	This study
pGEM-*hpten3*	PCR product of sequences encoding the C-terminal region (a.a. 186–403) of *hpten* cloned in vector pGEM-T Easy—Amp^r^	This study
pET-29c(+)	Expression vector for production of C-terminal His-tag fusions-Kan^r^	Novagen
pET29c-*hpten1*	pET29c overexpressing a hPTEN1-His_6_ fusion protein-Kan^r^	This study
pET29c-*hpten2*	pET29c overexpressing a hPTEN2-His_6_ fusion protein-Kan^r^	This study
pET29c-*hpten3*	pET29c overexpressing a hPTEN3-His_6_ fusion protein-Kan^r^	This study

**Table 4 molecules-29-02785-t004:** Sequences of the primers designed for PCR analysis.

Target Gene	Forward Primer (5′→3′)	Reverse Primer (5′→3′)	Amplicon Size (bp)
*PCNA*	GCCGAGATCTCAGCCATATT	ATGTACTTAGAGGTACAAAT	453
*CDKN1A (p21)*	GCAGACCAGCATGACAGATTTC	ATGTAGAGCGGGCCTTTGAG	127
*Bax*	ACCAAGAAGCTGAGCGAGTGTC	TGTCCAGCCCATGATGGTTC	255
*Bcl-2*	CCCTGTGGATGACTGAGTAC	GCATGTTGACTTCACTTGTG	211
*GAPDH*	GCACCGTCAAGGCTG	TGGTGAAGACGCCAG	138

## Data Availability

The experimental data are available upon request.

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
