# Peer review of "Carbon Nanotube-Mediated Delivery of PTEN Variants: In Vitro Antitumor Activity in Breast Cancer Cells"

_molecules, 2024, doi:10.3390/molecules29122785_

Round 1

Reviewer 1 Report

Comments and Suggestions for Authors

The manuscript entitled “Carbon nanotube-mediated delivery of PTEN variants: In vitro antitumor activity in breast cancer cells” by Papi RM et al. describes the construction and characterization of recombinant hPTEN1 and two truncated fragments-functionalized carbon nanotubes and evaluation of their antitumor activity in breast cancer cell lines.

The study is well structured with solid methodology and the results support the conclusions of the study. The data are novel with a clear biological significance.

Points

1.     The authors should provide indicative blots for the data presented in figs 1-5 and fig. 7.

2.     They should also mark with asterisks the statistically significant differences between values presented in figures 1-7.

Author Response

Dear Reviewer,

thank you for your comments. Please find our answers in the submitted file.

Reviewer 2 Report

Comments and Suggestions for Authors

Dear Editor

I have reviewed the manuscript entitled “Carbon nanotube-mediated delivery of PTEN variants: In vitro antitumor activity in breast cancer cells” My comments are as follows:

Author prepared PEGylated CNTs and bio functionalized multi wall CNTs by PTEN containing N-terminal and C-terminal tails for inhibiting the growth of T-47D, MCF-7 and ZR-75-1 human breast cancer cells. The functionalized multi wall CNTs were characterized by FTIR, TGA and Zeta potential. The manuscript was prepared well and interesting. However, there are some modification required before it as accepted.

1. The morphology of functionalized CNTs should be studied by FE-SEM.

2. Chemical structure of proposed bio functionalized CNTs should be provided.

3. In Figures S3 and S4, the line tone of graph should be darkening to understand clearly.

4. The concentration of CNTs 0 – 75 µg/mL were used in T-47D but 0 – 150 µg/mL against MCF-7 and ZR-75-1. Why?

Author Response

(The authors gave the same response as above.)

Reviewer 3 Report

Comments and Suggestions for Authors

In this article entitled (Carbon nanotube-mediated delivery of PTEN variants: In vitro antitumor activity in breast cancer cells), the manuscript is interesting however, some aspects should be better explored and explained.

Comments

Title

·        The title is suitable for the manuscript.

Abstract

·        All abbreviations in the abstract should be mentioned in full words.

·        The abstract is very concise and needs to be improved to allow the reader to understand the aim and value of this work. For example, you should mention the problem, aim, method, result, and conclusion without dividing the abstract.

Introduction

·        PTEN, in some cases written in italics, WHY? If there is no requirement please unify.

·        The reference cited in the introduction contains double brackets for example [[24,25]], line 74. Please revise the whole manuscript regarding this issue and you can use Zotero software to arrange the references.

·        You should cite the previous studies about using PTEN in treatment of any other type of cancer. If any.

Results and discussion

·        Please the table number in the manuscript. Table 1 in line 208 and also, and another table takes the same number in line 549.

·        The results and discussion part lack the interpretation with the previous findings.

·        Authors should represent the significant effect of CNTs-hPTEN in comparison with other groups (Control cells, cells incubated with different concentrations of CNTs-PEG)  in figures 2, 3, 4, 5, and 5.

·        Did the authors obtain ethical approval to do this study in the Human breast carcinoma cell line? If yes please add the ethical approval number.

Material and methods

·        For measuring the Zeta potential, what is the dilution % and the measuring temperature and angle?

·        Thermogravimetric analysis (TGA), in this part authors should mention which sample underwent thermogravimetric analysis and why. What is the reference pan in this experiment?

Conclusion

·        The conclusion is well-written

Comments on the Quality of English Language

Minor editing of English language required

Author Response

(The authors gave the same response as above.)

Reviewer 4 Report

Comments and Suggestions for Authors

Thanks for your kind efforts. The results supported well your hypothesis. Modifications of CNT might need to be more detailed with characterization.

Comments on the Quality of English Language

Just have a second round for punctuation and the use of in, on. etc

Author Response

(The authors gave the same response as above.)

Round 2

Reviewer 2 Report

Comments and Suggestions for Authors

The manuscript was revised and it can be accepted for publication in Molecules in its present form

Reviewer 3 Report

Comments and Suggestions for Authors

The authors improved the quality of their manuscript, which now can be considered for publication.